# Identifying the perceived local properties of networks reconstructed from biased random walks

**Lucas Guerreiro**[1], **Filipi Nascimento Silva**[2]*, **Diego Raphael Amancio**[1]

**1** Institute of Mathematics and Computer Science – USP, Avenida Trabalhador São-carlense, São Carlos, SP, Brazil, **2** Indiana University Network Science Institute, Bloomington, Indiana, United States of America

* filsilva@iu.edu

**Data Availability Statement:** The data can be found from the cited sources in the paper. No new datasets are being introduced in this paper.

**Funding:** Diego R. Amancio acknowledges financial support from CNPq (grant no. 311074/2021-9) and

## Abstract

Many real-world systems give rise to a time series of symbols. The elements in a sequence can be generated by agents walking over a networked space so that whenever a node is visited the corresponding symbol is generated. In many situations the underlying network is hidden, and one aims to recover its original structure and/or properties. For example, when analyzing texts, the underlying network structure generating a particular sequence of words is not available. In this paper, we analyze whether one can recover the underlying local properties of networks generating sequences of symbols for different combinations of random walks and network topologies. We found that the reconstruction performance is influenced by the bias of the agent dynamics. When the walker is biased toward high-degree neighbors, the best performance was obtained for most of the network models and properties. Surprisingly, this same effect is not observed for the clustering coefficient and eccentric, even when large sequences are considered. We also found that the true self-avoiding displayed similar performance as the one preferring highly-connected nodes, with the advantage of yielding competitive performance to recover the clustering coefficient. Our results may have implications for the construction and interpretation of networks generated from sequences.

## 1 Introduction

Many real-world phenomena are characterized by discrete series of events or decisions happening in succession [1, 2]. This includes how users navigate through websites or social media, how language is written and spoken, music, city navigation, and even people's everyday decisions. In most cases, however, only a limited amount of information is available to infer the rules and the mechanisms driving the generative processes behind these phenomena. For instance, from the perspective of a social media user, the observed content may be limited to their own interests, political positions, friends' preferences and what is being suggested by a recommendation algorithm. Such content normally only constitutes a small fragment of what is present in the complete social media platform. These aspects are often linked with the emergence of biases leading to polarization, formation of echo chambers, and other social

São Paulo Research Foundation (FAPESP grant no. 2020/06271-0). This research was supported in part by Lilly Endowment, Inc., through its support for the Indiana University Pervasive Technology Institute. The funders had no role in study design, data collection and analysis, decision to publish, or preparation of the manuscript.

**Competing interests:** The authors have declared that no competing interests exist.

phenomena like the friendship paradox [3]. In another example, because of limited individuals' capacity, resources and available personnel, scientists or research groups adopt different strategies to choose the focus of their research among all the possible problems. Such a strategy could favor exploitation over exploration (or vice versa), a decision that could potentially impact the collective discovery process [4]. These examples raise the question on how well aspects of the inherent (generative) process are truly recovered through limited or biased information.

Since many complex systems have been successfully represented by networks (i.e., by the intricate relationships among their components) [5–8], it is possible to study the aforementioned question in terms of how well the characterization of such structures changes according to the limited information observed through certain dynamics. In particular, for the case of networks, different behaviors of dynamics can be simulated through random walk heuristics. In such systems, an agent performs a walk in a network and reconstructs it based on the set of visited nodes and edges. This process has been investigated, in particular for the case of the knowledge acquisition process, in which pieces of knowledge are learned by agents walking across a network representing knowledge. Previous works [9] have found that it is possible to determine characteristics of the inherent model by only looking at features of the partially reconstructed networks. This indicates that different combinations of network topology and dynamics can lead to potentially different observed features in the generated sequences. In this work, we address the problem of checking how similar are the observed features of partially reconstructed networks compared to the original structure.

We approach the problem of reconstructing networks from limited information by employing different types of random walks performed by non-interacting agents. Each agent simulates an individual with limited information and stores a subgraph of the original network reconstructed by co-adjacency. Fig 1 shows an example network in which a random walk was performed starting at node A. This simulates, for instance, a user in social media navigating across different profiles. Given the user's limited information, they may think that node A is the one with most connections in the network, in contrast to the correct answer: C. This is because the agent visited A's neighborhood, while B's and C's neighbors were not. Other properties such as clustering coefficient and centrality measures are also not correctly recovered

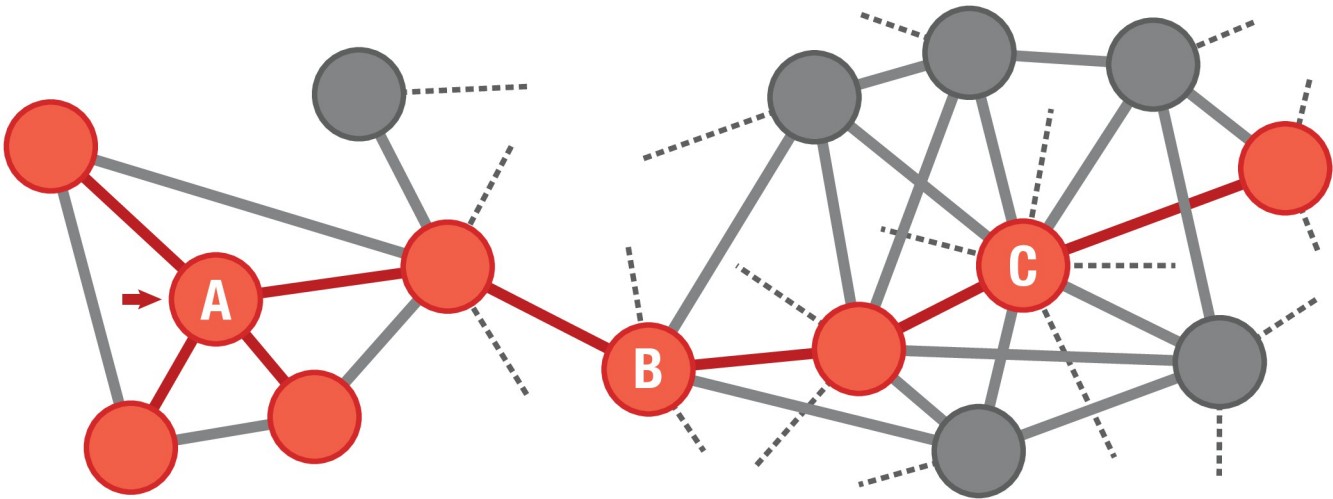

**Fig 1. Example of a subgraph (highlighted in red) representing the limited information observed by a random walk starting at node A.**

through this walk. The potential to recover network properties may also depend on the network topology itself. For instance, an irregular network with heterogeneous degrees and high density may be more challenging to navigate than a regular network, since in the first case, hubs may be visited more frequently, potentially leaving regions of low degree poorly explored. As a consequence, the recovered network characteristics may be different and biased compared to those from the original network.

In this work, we study how well network characteristics—such as node degree, clustering coefficient, etc—can be recovered from reconstructions based on finite sequences. We explore the effects of different strategies to generate the sequences, including biased [9] and true self-avoiding [10, 11] random walks. First, we generate sequences of nodes based on the progression of visited nodes given by an agent dynamics. Next, the sentences are used to reconstruct independently a network based on co-adjacency. Network properties for both the original and reconstructed networks are obtained and compared via Pearson or Spearman correlations. We also vary the length of the sequences to simulate different levels of limited information. For this analysis, we considered real-world networks and realizations of traditional network models in addition to a community-based model, the LFR [12].

Our results indicate that the choice of dynamics employed to generate the sequences has an influence on the correlation values between the recovered and original network properties. The reconstruction performance depends, for instance, if the dynamics are biased by node degree. When highly connected nodes are preferable to be visited (RWD), we achieve the best performance in recovering network properties for most of the considered networks and properties, with the exception of clustering coefficient and eccentricity. In those cases, even by considering the long sequences, it still reaches low values of correlation. On the other hand, for the case that the random walk dynamics avoids highly connected nodes (RWID), we see the worst performance among the considered dynamics. However, it is able to recover the clustering coefficient with similar performance as other dynamics. In addition to that, we explore three other types of random walks, the unbiased random walk (RW) and two self-avoiding strategies, known as true self-avoiding walk [10, 11], one based on edges, which avoids passing through already visited edges (TSAW-edge), and another based on nodes (TSAW-node). TSAW-edge displayed similar performance as the RWD approach, but with no problems in recovering the clustering coefficient. We discuss these results in detail and the potential implications in Section 4.

Finally, we also check if the community structure can be recovered from the partial information stored in sequences. This is accomplished by comparing the detected communities' membership of the original networks (or planted for the LFR models) with those from the reconstructed versions. The results seem to depend strongly on the network topology, with mixed patterns across different mixing coefficients of the LFR and real networks. Nonetheless, TSAW-edge and RW display the best performance in that task.

This work is organized as follows. In Section 2 we present and discuss the related works. Section 3 describes the adopted methodology in order to generate networks, perform random walks on the topologies, reconstruct partial networks and how we have analyzed and compared the properties in these networks. The results are reported and discussed in Section 4. Finally, in Section 5, we present the general conclusions and future works.

## 2 Related works

The process of random walkers exploring complex network topologies has already been studied by several works [2, 13–15]. In the context of knowledge acquisition, the sequence of visited nodes in random walks is to recover the set of nodes in the network [2, 14]. In [2], the

authors investigated how different agents walking over the network can reconstruct the network topology. In the proposed multi-agent random walk, the true self-avoiding and Lévy flight-based dynamics outperformed other walk strategies in terms of efficiency in discovering new nodes. Surprisingly, the study also showed that fine-tuning the parameters controlling the agent dynamics had little effect on the global knowledge acquisition performance.

The study conducted in [13] focused on the knowledge acquisition task when several network topologies and agent dynamics are used in a single-agent context. This study found that the true self-avoiding dynamics had the best performance over different settings in discovering nodes in the network. The degree-biased had the slowest learning curve in the experiments. The study has also demonstrated that higher average degrees provide a faster learning rate.

While several studies focused on the knowledge (nodes) acquisition problem [14, 15], the study conducted in [9] used a machine learning approach to recover both the network topology and agent dynamics generating a sequence of symbols. To train the supervised classifiers, sequences of visited nodes were mapped into (reconstructed) networks via the co-occurrence strategy. Then, six different network properties were used to create features describing the observed reconstructed networks. Sixteen different combinations of network topology and agent dynamics were considered to generate sequences. The study revealed that it is possible to recover both the topology and dynamics with high accuracy, provided that the sequence (i.e the random walk) length is not too short. The accuracy of identification increased with the observed sequence length. When less than 20% of the whole network was discovered, both the topology and dynamics were recovered with an accuracy higher than 86% in a supervised classification scenario with 16 classes.

In [16], the authors analyzed how network properties (e.g. average degree) evolve as the network sample size grows. If a network property is unstable for all sample sizes then it does not represent the network very well; however, if the property does not change as the sample size grows, then the property is considered a good representation of the network. The main contribution of this work is therefore a methodology to quantify if any network property is robust regarding the network size used in the experiments. In networks formed from sequences, it means that unstable properties may vary depending on the sequence length used to form the networks. This means that when recovering local properties, the original value of the property may only be recovered if the sample and original network sizes are consistent. However, one may still find a correlation between values observed in sampled and original networks for unstable metrics.

The study conducted in [7] investigated a teaching-learning perspective using complex networks. In the adopted representation, facts are graph nodes and the relationship or underlying connections between two facts are represented by edges. The study aimed to probe how students learn contents from linear algebra textbooks by considering the nodes exploration process simulating the human memory characteristics during the learning process. Among the main findings, the authors reported that human memory limitation plays a special role in long-term information retention effectiveness and problem-solving creativity.

The relationship between knowledge representation and complex networks has also been studied elsewhere. In [14], knowledge is acquired when nodes and edges are visited by random walkers. Different from other approaches, the experiments considered free and conditional transition edges. While the former is commonly used in most of the works, the latter allows new nodes to be accessed only when certain criteria are met. In this case, the main criteria consist in visiting a subset of nodes in order to make a new node accessible. The author analyzed the knowledge acquisition performance via hierarchical complex networks [17], which are

explored via traditional random walks and variations biased toward new links. The study showed that the biased random walks are slower to acquire knowledge in the conditional exploration scenario.

While most of the related works tried to recover the set of nodes or identify the dynamics and topology generating a sequence, here we focused on a different network perspective. We studied if the properties of the reconstructed networks are consistent with the ones of the original networks. This is a relevant topic since in many scenarios one does not have access to the original networks generating a sequence of symbols.

## 3 Methodology

In this section, we discuss the proposed methodology. The main purpose of this paper is to analyze whether the local properties of reconstructed and original networks are correlated. The methodology can be divided into the following 4 main steps, which are summarized below. The steps are also illustrated in Fig 2.

- *Original networks*: here we used network models to represent different network topologies. Examples of models include random and geographical networks. We also used examples of real-world networks modeling e.g. social and biological complex systems.

- *Network dynamics*: in many real-world situations, network data is only available as a sequence of symbols [18]. Sequences can be generated by an agent walking over the network via different rules.

- *Network reconstruction and properties extraction*: the observed sequence generated in the previous step is used to reconstruct. Several properties of the obtained networks are then extracted.

- *Correlation analysis*: the properties of the original and reconstructed networks are compared. The properties are compared in terms of network metrics (e.g. clustering coefficient) and, in modular networks, the partitions representing the network communities are compared.

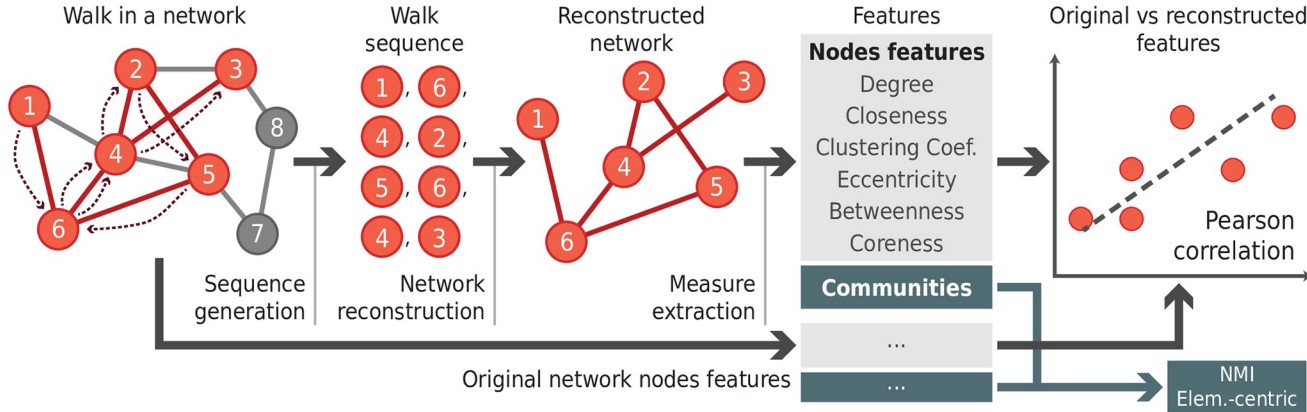

**Fig 2. Schematics of the methodology.** First, we look into the original network and annotate the concerned properties for each node. In the next step, we iterate over the network with the desired dynamics in order to generate a sequence of symbols. In the third step, we reconstruct the network using the discovered nodes and edges, and we annotate the node's properties in this reconstructed network. Finally, we build correlations among the reconstructed and original nodes by comparing each node *i* in the reconstructed network to its respective node in the original network.

## 3.1 Original networks

We considered the most common and diverse topology models in the literature [19]. Similar to previous studies [9], our analysis focused on networks with $N = 5,000$ nodes and average degree $\langle k \rangle = 4$. The following models were used here:

- *Erdős-Rényi (ER)*: this is the traditional random network model. The probability $\pi$ of two nodes being linked by an edge is constant [20, 21].

- *Barabási-Albert (BA)*: this model implements a scale-free topology [22]. Different from ER networks, the BA model is based on a growth model, since at each step new nodes are included in the network. The probability $\pi_{ij}$ of a new node $i$ to connect to an old node $j$ with $k_j$ links is proportional to $k_j$:

$$\pi_{ij} = \frac{k_i}{\sum_j k_j}. \tag{1}$$

- *Waxman (Wax)*: this topology is an implementation of a geographic network. The first step consists in randomly placing each node in a two-dimensional plane. A link between two nodes $i$ and $j$ is provided by a probability formulation that decays exponentially with the geographical distance between the nodes [13, 23].

- *Modular Networks (LFR)*: this model [12] creates networks with nodes clustered into network communities. The main parameters used to construct modular networks are the number of communities ($n_C$), the exponent for the degree sequence in the network ($t_1$), the minus exponent for the community size distribution ($t_2$), and the mixing parameter ($\mu$), which measures how well defined communities are. Lower values of mixing value lead to well-defined network communities. We used the following parameters to construct the networks: $n_C = 5$, $t_1 = 3$, $t_2 = 0$ and $\mu = \{0.05, 0.2, 0.8\}$. Similar values have been used in related works [2, 9, 13],

We also conducted our experiments in real networks modeling diverse complex systems, the selected networks description as well as their download links are:

- *Facebook*: this network comprises social relationships among Facebook employees. The network comprises 320 nodes [24] and is available at [25].

- *Power Grid*: this is a classic geographical network modeling the US Western States Power Grid. Nodes represent transforms or power relay points, while edges are power lines. The network comprises 4,941 nodes [26] and is available at [25].

- *Econ-Poli*: we have used Economics Poli network which contains 3,915 nodes and presents behavior on interconnected economic agents [27]. The network is available at [27].

- *Web-EPA*: we also have used the Web-EPA network with the size of 4,271 nodes and implements information in web level for hyperlinks across the internet that link to the www.epa.gov website [27]. The network is available at [27].

- *Bio (DM-CX)*: The Bio (DM-CX) is a biological real network that represents co-occurrences on *Drosophila melanogaster* fly pairs of genes acquired from the FlyNet repository. The network comprises 4,032 nodes [27, 28]. The network is available at [27].

- *Bio (AI Interactions)*: we have explored the AI Interactions dataset which presents a biological real network with *Arabidopsis Interactome* map of protein-protein interactions. The largest component of this network comprises 4,519 nodes [29]. The network is available at [30].

- *socfb-JohnsHopkins55*: this is a social network representing Facebook connections inside the Johns Hopkins community. The network comprises 5,180 nodes and is available at [27].

Since the size of the networks can influence the coverage speed of random walks, we opted to select networks of similar size. Except for the Facebook network, all the choosen networks have about 4000-5000 nodes.

## 3.2 Network dynamics

Agent dynamics have been used in a wide variety of network-based studies, including epidemic spreading and knowledge acquisition analysis [2, 13, 15]. In this work, agent dynamics are used to explore the network and generate a sequence of visited nodes. The sequence of nodes is assumed to be the information available for network reconstruction. We have used 5 well-known walks:

- *Traditional Random Walk (RW)*: in the traditional random walk the agent selects the next node to visit randomly among its neighbors. The probability of the agent moving from node $i$ to node $j$ is $p_{ij} = 1/k_i$, where $k_i$ represents the degree of $i$-th node.

- *Degree-biased Random Walk (RWD)*: this random walk considers the degree of the neighbor nodes when selecting the next node to be visited by the agent. The probability of visiting a node is proportional to its degree:

$$p_{ij} = \frac{k_j}{\sum_{l \in \Gamma_i} k_l}, \tag{2}$$

where $\Gamma_i$ is the set comprising the neighbors of $i$.

- *Inverse of the Degree-biased Random Walk (RWID)*: Similar to the RWD dynamics, the RWID walk uses the degree of the neighborhood when defining the probabilities. However, in this dynamics, the agent prefers to visit the nodes with smaller degrees, i.e.

$$p_{ij} = \frac{k_j^{-1}}{\sum_{l \in \Gamma_i} k_l^{-1}}. \tag{3}$$

- *True Self-avoiding Random Walk on nodes (TSAW-node)*: in this random walk, the agent avoids the nodes that were already visited [10, 11]. Therefore, in this network, the nodes not yet visited are preferred to be visited, which works in favor of the network exploration. Let $f_j$ be the frequency that $j$ has been visited. The mechanism to avoid already visited nodes is encoded according to:

$$p_j = \frac{e^{-\lambda f_j}}{\sum_{l \in \Gamma_i} e^{-\lambda f_l}}. \tag{4}$$

- *True Self-avoiding Random Walk on edges (TSAW-edge)*: similarly to the traditional TSAW dynamics, in this random walk there is an avoiding bias, however in this implementation, the agent avoids edges previously visited instead of nodes, as implemented in works related to network exploration [2, 13]. The probability transition is computed as

$$p_{ij} = \frac{e^{-\lambda f_{ij}}}{\sum_{l \in \Gamma_i} e^{-\lambda f_{il}}}, \tag{5}$$

In both versions of the true self-avoiding random walks, we are using $\lambda = \ln 2$, as in related works [2, 9, 13].

### 3.3 Network reconstruction and properties extraction

The sequences generated by the random walks are used to reconstruct the networks. In our experiments, we probed how the sequence length affects the properties of the reconstructed networks. We investigated the results for the following set of sequence length $w$: {100, 200, 400, 500, 600, 800, 1000, 2000, 5000, 20000, 50000}. The reconstruction is performed by recreating the edges observed in the sequence. This procedure is equivalent to the co-occurrence approach usually employed in network analysis, where two symbols are linked whenever they are adjacent in the sequence. When analyzing texts using network science, the co-occurrence approach is widely employed [6, 31, 32]. For each combination of network topology, agent dynamics and walk size, we considered 20 sequence realizations.

Once the network is reconstructed, our aim is to analyze if relevant properties can be recovered. To characterize the networks, we used well-known network metrics, including local, quasi-local and global metrics. We computed the degree, clustering coefficient, closeness, betweenness eccentricity and coreness centrality of the networks [19, 33]. All metrics are defined for unweighted and undirected networks. For networks with modular structure, we also detect the communities using the Leiden method [34].

### 3.4 Correlation analysis

To assess how similar the structural properties are preserved by the reconstruction process, we employed the Pearson and Spearman correlations between structural properties of the original and reconstructed networks. More specifically, for each node $i^{(R)}$ of the reconstructed networks we measure a structural feature $\mu(i^{(R)})$ (e.g. degree or clustering coefficient). The same property is also measured for the corresponding node in the original network, i.e. $\mu(i^{(O)})$. Finally, we measured the correlation between $\mu(i^{(R)})$ and $\mu(i^{(O)})$, for all nodes of the reconstructed network.

Since we are also interested in the modular structure of networks, we also compared the similarity of partitions in the original and reconstructed network via normalized mutual information (NMI) [19, 35] and adjusted rand index (ARI) [36, 37]. Higher values of normalized mutual information mean that the partitions of the original and reconstructed network are similar.

The steps taken in our framework and that were described in this section are summarized in Algorithm 1. The code can be found on GitHub at: https://github.com/lucasguerreiro/localproperties.

To mitigate the inherent randomness associated with the starting nodes in random walks, we conducted multiple iterations of each walk for every network configuration (20 repetitions for each configuration), thereby ensuring that the Pearson correlation values presented are robust averages.

**Algorithm 1** Framework to explore networks ($N_o$), reconstruct subnetworks ($N_r$) and calculate correlations between $N_r$ and $N_o$

```
function RECONSTRUCT(r)
    N_r ← ∅
    lastnode ← ∅
    for n ∈ r do
        if n ∉ N_r then
            N_r ← N_r∪n        ▷ add node n to N_r
```

```
      end if
      N_r ← N_r∪(lastnode, n)        ▷ add edge (n,lastnode) to N_r
    end for
end function and return N_r
for p ∈ w do
    r ← walk(N, d, p)        ▷ perform dynamics d in network N with length
p
    N_r ← RECONSTRUCT(r)
    μ(i(R)) ← getproperty(N_r)       ▷ calculate property of reconstructed
network
end for
μ(i(O)) ← getproperty(N)       ▷ calculate property of original network
C ← correlation(μ(i(O)), μ(i(R)))         ▷ calculate correlation
```

## 4 Results and discussions

In Section 4.1, we analyze if the local properties of the original networks are preserved for distinct biased random walks. In Section 4.2, the efficiency in recovering local properties is analyzed in the context of the knowledge acquisition task [2].

### 4.1 Efficiency in recovering the original properties

In our first analysis, we intended to probe whether the properties of the reconstructed networks are consistent with the properties of the original ones, according to the procedure described in Fig 2. The scatter plot of the node degree observed in original and reconstructed networks is shown in Fig 3 for the five agent dynamics in the LFR network (with mixing parameter $m = 0.05$). Each column represents a different sequence length (chosen proportionally to the original network length), while lines are different agent dynamics. For each subpanel, we show in the x- and y-axis the node degree observed in the reconstructed and original networks, respectively. We also show in each subpanel the Pearson and Spearman correlations ($C_p$ and $C_s$, respectively).

We notice that for small sample sizes ($w = 100$ and $w = 500$) the node degree property is not well represented since the agent did not acquire enough information to create an accurate representation of the original network. This might be an explanation of previous results showing that networks reconstructed via very short walks are not consistent with their original topological nature [9]. Interestingly, larger sample sizes not necessarily imply high correlation values. Considering the RWID agent dynamics and $w = 5,000$ steps, the Pearson correlation, $C_p$, reaches only 0.24, even though higher values were observed for the other considered dynamics. Considering the same walk length, we found $C_p = 0.88$ for the RWD walk. This means that the node degree is consistent (i.e. linearly correlated) with the ones observed in the original networks.

For large values of $w$, i.e. long sequences, one should expect that most of the network structure (nodes and links) is retrieved by the agents [2]. Therefore, the correlations should reach high values. In fact, this is observed in the TSAW Edge dynamics ($C_p = 1$). This means that this particular TSAW is not only efficient in knowledge acquisition, but it also captures the local connectivity [2]. RWD outperformed RW in this particular network. Finally, it is clear that after $5,000$ steps, the RWID walk may recover relevant information regarding node connectivity but it does not perform as well as the other dynamics.

While in Fig 3 we focused on a scatter-plot analysis of a single network model, the scatter plot for other network models are similar to the one shown for the other LFR networks (results not shown). The behavior of the correlations in different agent dynamics and network topologies is shown in Fig 4. The figure illustrates the evolution of the Pearson correlation for distinct

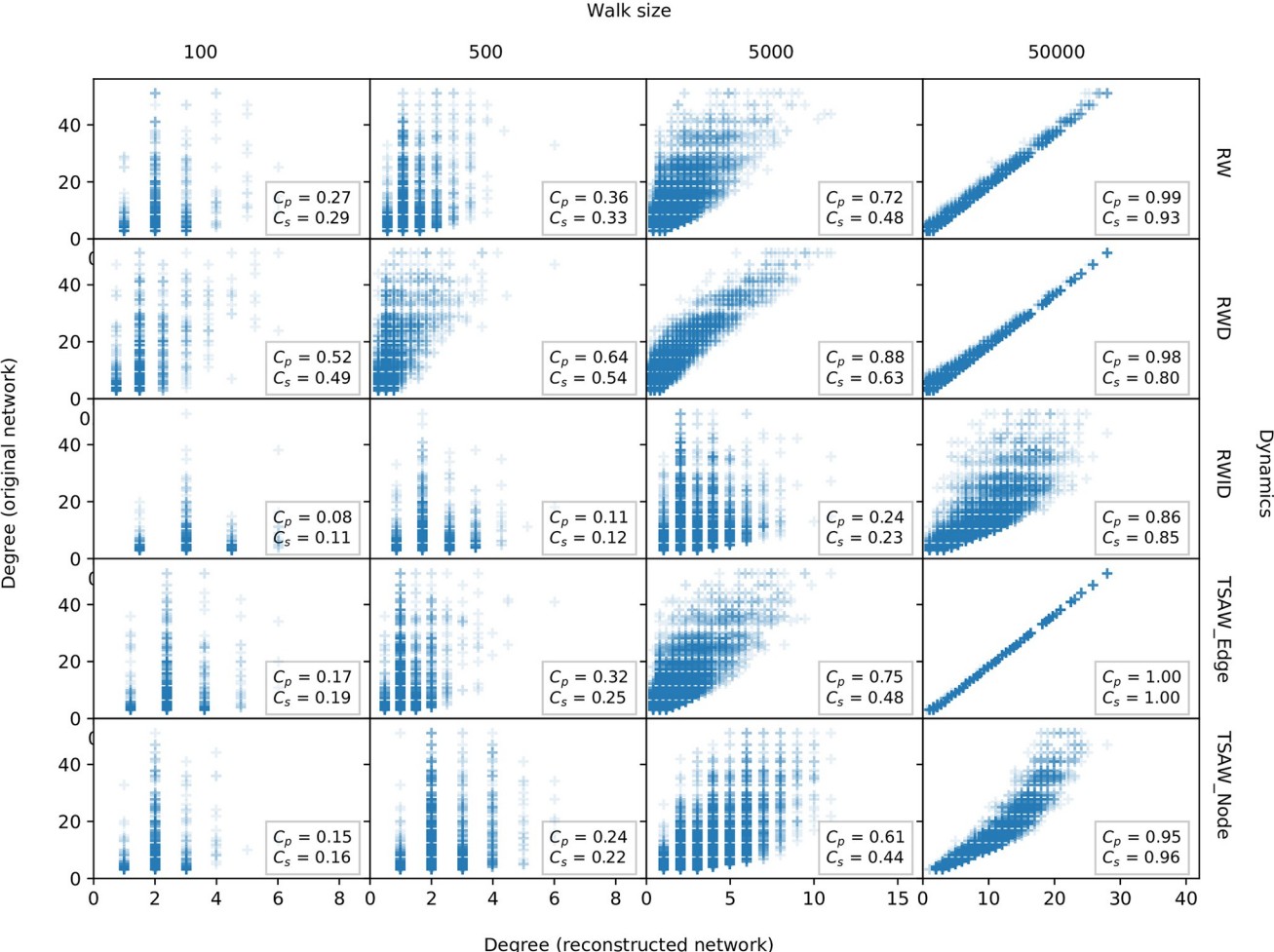

**Fig 3. Scatter-plot of node degree obtained in original vs. reconstructed LFR networks (with mixing parameter _m_ = 0.05).** Each subpanel corresponds to a different configuration of agent dynamics and walks length (_w_). The walk length is distributed between 100 and 50,000 steps.

sequence lengths. We also show the NMI and ARI, comparing the structure of communities found in the LFR original and reconstructed networks.

The results in Fig 4 reveal that the RWID dynamics displayed the lowest correlation values in almost all scenarios. This is especially true in networks where hubs play a prominent role, as is the case of BA and LFR networks. Therefore, avoiding hubs in networks where hubs are relevant causes a distortion in network metrics observed in reconstructed networks. Conversely, the RWD dynamics displayed competitive performance, even in networks with no evident presence of hubs (see e.g. degree in ER networks). The efficiency of RWD is more evident in BA and LFR networks, even in short sequences. The RW dynamics displayed a performance that is similar to the TSAW Edge walk. A major difference was found only for particular metrics, e.g. the eccentricity in ER networks for long sequences. Interestingly, we found that major differences in performance can be found for different versions of the TSAW walk. This is the case of the betweenness in BA networks. For walk lengths larger than 2, 000 steps, the true self-avoiding rule applied on edges turned out to be more efficient than the same rule applied on nodes.

The efficiency of metrics recovery has a minor dependency on the walking strategy when considering the clustering coefficient, coreness and the NMI and ARI metrics. For the

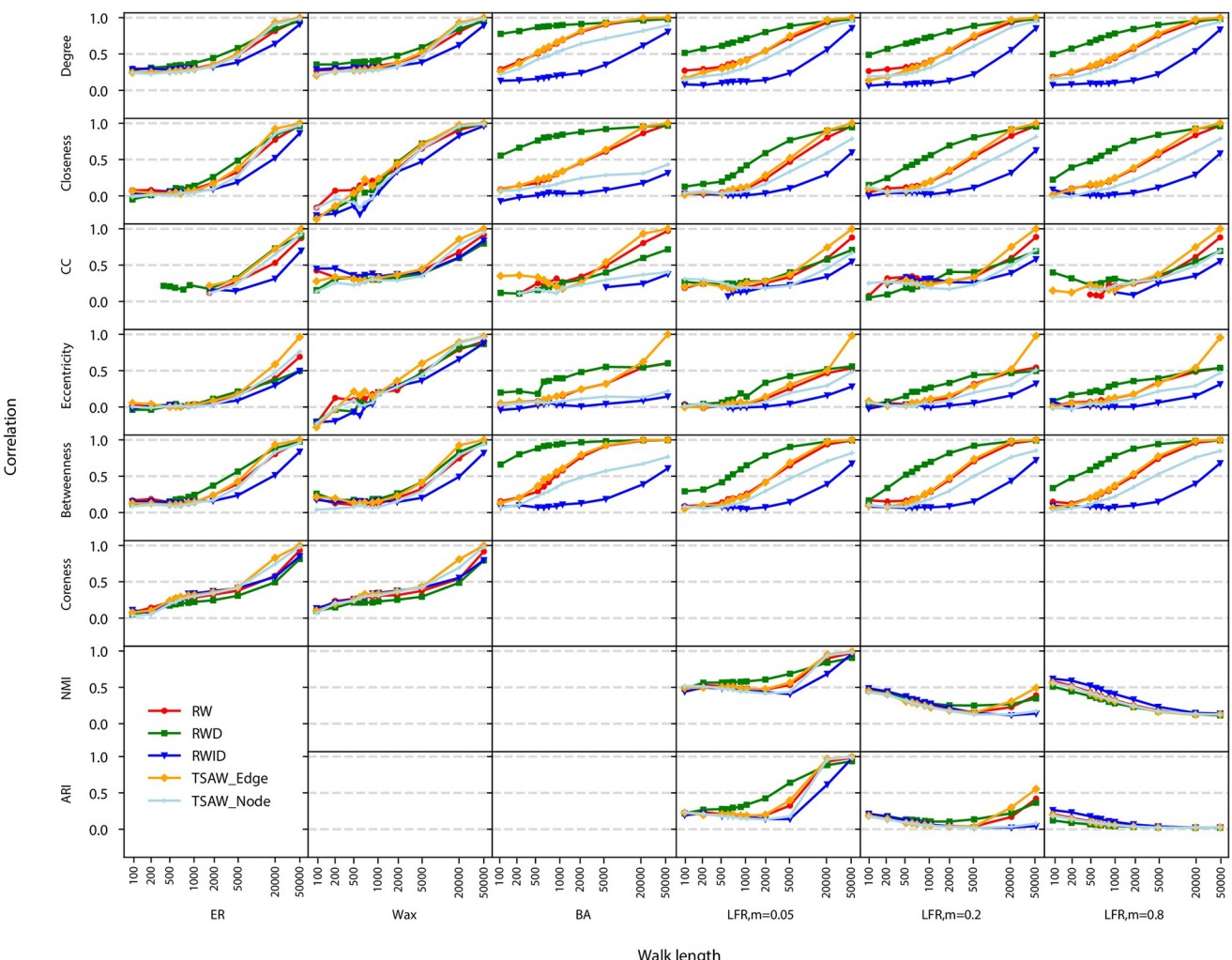

**Fig 4. Evolution of the efficacy in recovering network metrics in reconstructed networks.** The x-axis represents the walk length used to generate the sequence, and the y-axis is the Pearson correlation for local metrics obtained in the original vs. reconstructed networks. The last columns are the NMI and ARI, which are used to compare the partitions in networks with community structure. While the coreness is not defined in BA and LFR networks, the NMI and ARI are computed only for networks with community structure.

particular case of networks with community structure (i.e. LFR networks), we noticed that there is no evident differences in performance in networks with high values of mixing parameter. The differences in performance are only evident for well-defined communities, i.e. for networks with mixing parameter lower than 0.20. However, the NMI reaches roughly 0.50 even after long walks. In well-defined communities (mixing parameter = 0.05), we found that the structure can indeed be recovered; also, we noticed that the ARI metric did not differ much from the NMI calculations, presenting corresponding results. However, a large number of steps is still required to achieve high performance.

In Fig 5 we show the efficiency of the recovery for real-world networks obtained from systems of different disciplines (as described in Section 3.1). We observe the same overall behavior for both RWD and TSAW Edge dynamics. In most cases, the RWD outperforms other approaches for short sequences, while the TSAW dynamics performs better when longer sequences are considered. We can also notice that, again, the RWID dynamics had the worst

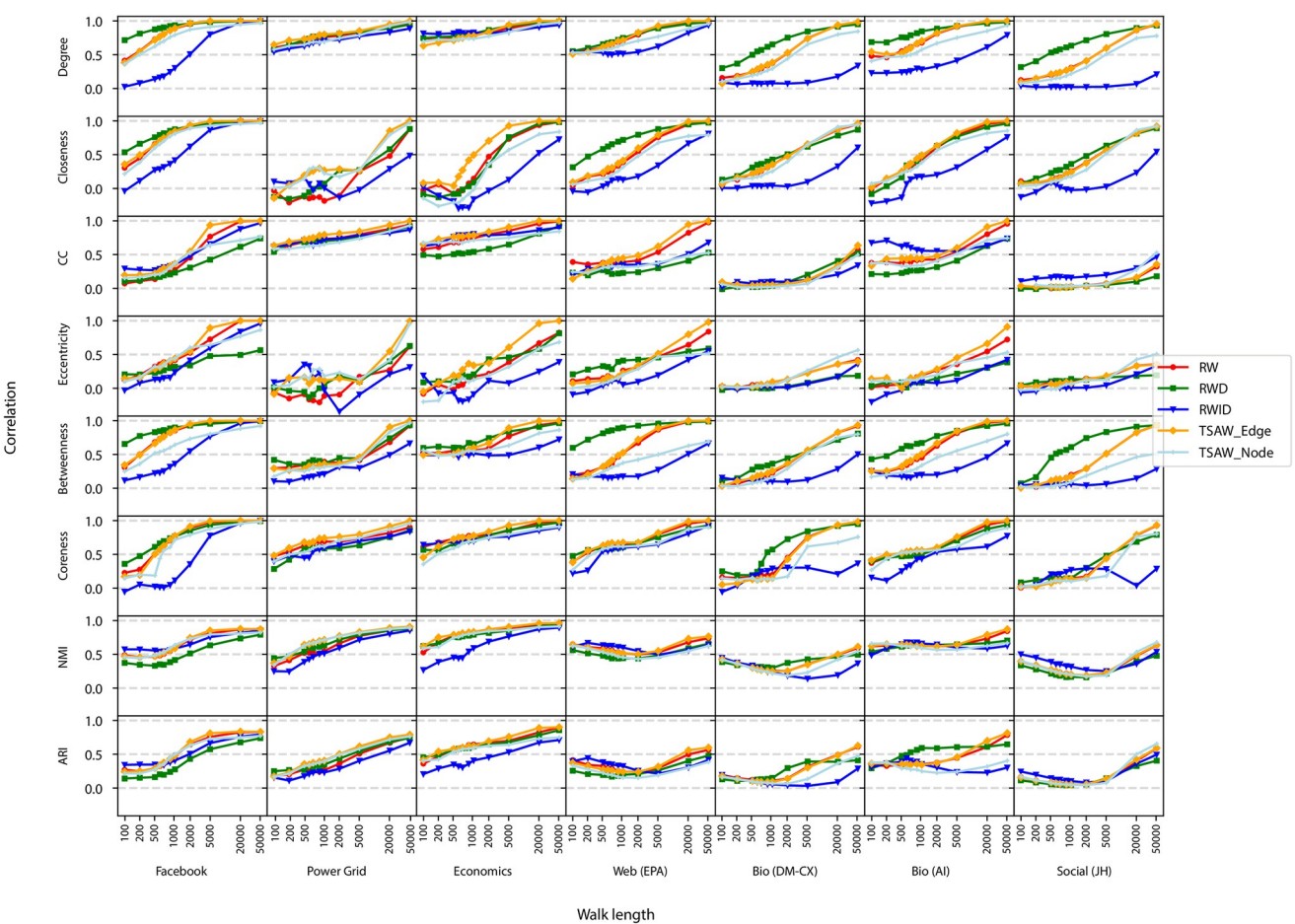

**Fig 5. Pearson correlation for the selected real networks.** Each data point represents the correlation of the same nodes for the original and reconstructed network from sequences acquired by dynamics for a given walk length, $w$ ($100 \leq w \leq 50000$). The last columns represent the average NMI and ARI of each model obtained with the Leiden algorithm.

comparable performance in terms of correlation over the original network for most properties.

When comparing the efficiency of properties recovery across different networks, the Economics and Power Grid networks have almost all of the considered properties recovered with high correlation for sequences comprising more than 2,000 nodes. Conversely, for some properties in both Bio (DM-CX) and Social (JH) networks, 50,000 steps were not sufficient to recover the original metrics with high efficiency. This is the case of the clustering coefficient, eccentricity and coreness. Concerning the different network properties, the community structure could be recovered with efficiency only for three networks (according to the NMI and ARI metrics). Interestingly, we can see that neither walk outperformed the others substantially regarding the recovery of network structure.

## 4.2 Efficiency in recovering network properties and knowledge acquisition

While in the previous section we focused on analyzing the recovery of network properties, here we also consider the knowledge acquisition performance as an additional feature of the random walks [13]. In the *knowledge acquisition* task, each node is considered as a piece of

knowledge, and the performance metric corresponds to the fraction of the total number of nodes that have been discovered in the reconstructed network in comparison with the original network [13]. In this context, we analyze whether the reconstructed network is a good representation of the original ones in a twofold fashion: (i) the correlation of the properties of the discovered nodes; (ii) the computation of how many nodes from the original networks have been recovered.

In Fig 6, in the x-axis, each point represents the knowledge acquired by the walkers for a given sequence length $L$, i.e. we show the fraction of unique nodes discovered for that sequence length. In the y-axis, we show the correlation of properties obtained for the same value of $L$, according to the methodology described in Section 3.3. One may notice that, in most scenarios, the RWD dynamics improved the recovery correlation as more nodes are discovered. In particular scenarios, high correlation values are reached in the first steps of the walk, however, many more steps are required to discover a significant portion of the network (see e.g. the closeness metric for the BA network). We also note that the knowledge acquisition and correlation performance may increase with a similar speed—this is the case e.g. of the clustering

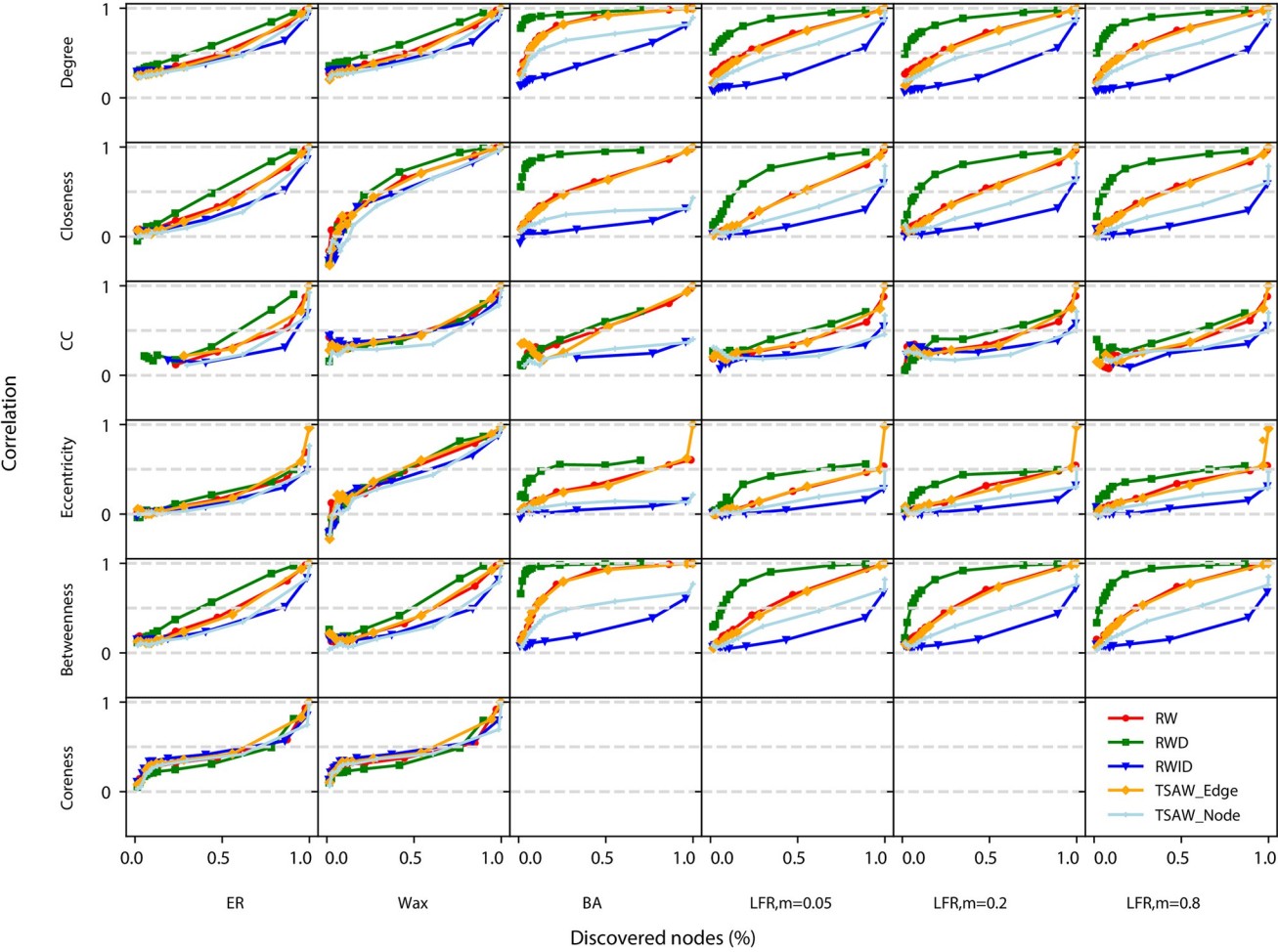

**Fig 6. Efficiency in recovering network metrics in for the network models.** Each data point represents the correlation of local network metrics between the original and reconstructed network from sequences acquired by dynamics for a given walk length, where the x-axis represents the knowledge acquired for each walk length ($w$).

coefficient in BA networks). As for the RWID dynamics, we observe that, in many cases, even when a large portion of the network is discovered, a low correlation is found (see e.g. the closeness centrality in BA networks). As for the TSAW, in most cases, the edge-based version presented a higher correlation when the same amount of nodes was discovered. This is evident, for example, when recovering the betweenness in BA networks. When half of the network is discovered, the edge-based version presents an almost perfect correlation, while the node-based version only achieves a correlation value close to 0.50.

We have also analyzed both correlation and knowledge acquisition relationships in real-world networks. The results are shown in Fig 7. In the Facebook network, for a fixed amount of discovered nodes, the highest correlation is mostly achieved with RWD dynamics. Both clustering and eccentricity metrics This result is compatible with the behavior of BA networks. Surprisingly, in the Power Grid, Economics and BIO (AI) networks, there is no evident difference in the behavior observed for distinct random walks for most of the considered metrics. The RWD again seems to provide the highest values of correlation when the same number of nodes is discovered for shortest paths-dependent metrics (closeness and betweenness) in the Web network. Finally, we note that RWD also achieved the highest accuracy for the degree, closeness and betweenness in the social network. Finally, in almost all metrics and networks, we again observed that the TSAW-edge strategy is more efficient in recovering nodes' properties than the nodes-based counterpart.

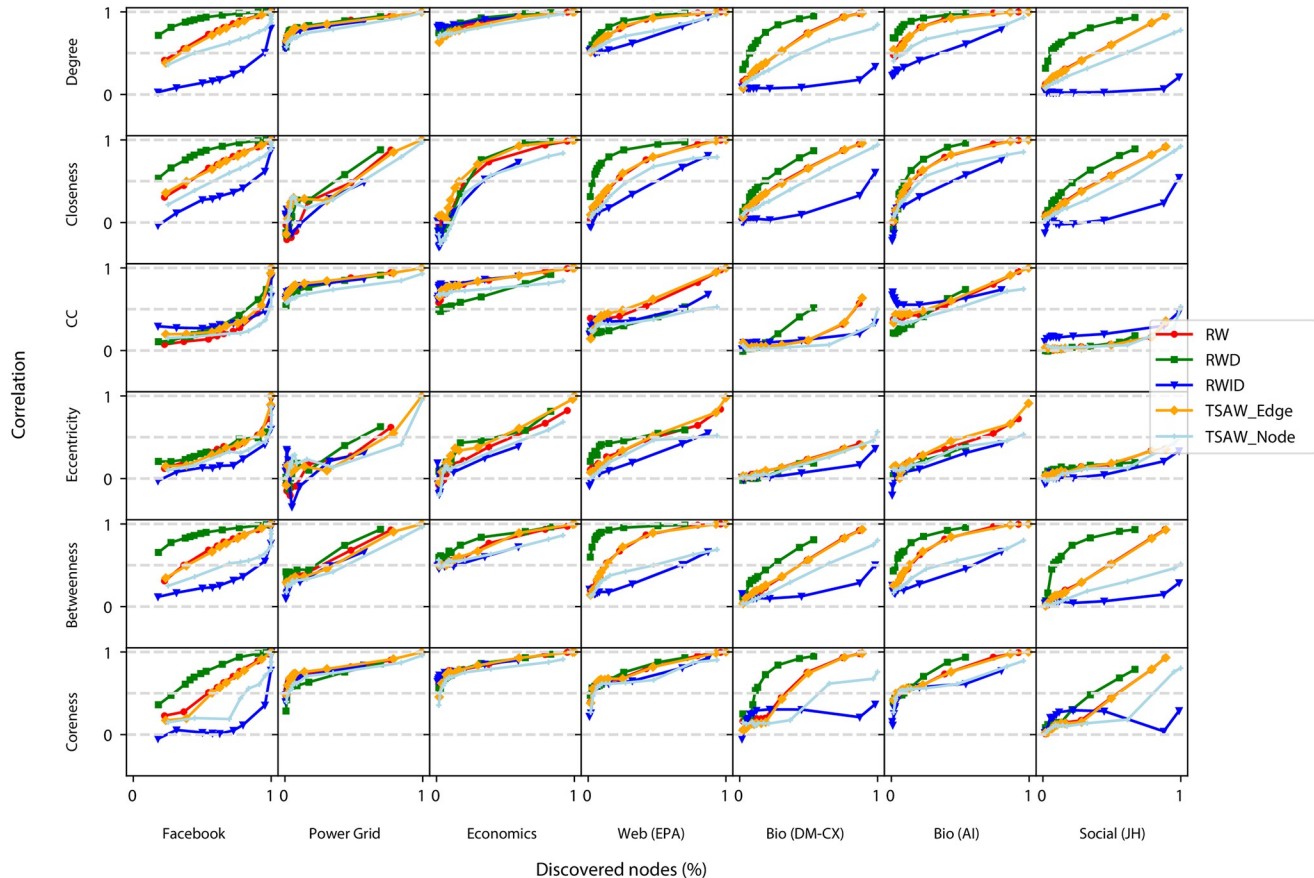

**Fig 7. Pearson correlation for the selected real networks.** Each data point represents the correlation of the same nodes for the original and reconstructed network from sequences acquired by dynamics for a given walk length, where the x-axis represents the knowledge acquired for each walk length, $w$, where $100 \leq w \leq 50000$.

All in all, the results indicate that the RWD agent dynamics performs equally or better than the other walk dynamics when recovering local metrics when the same fraction of nodes is discovered. We should keep in mind, however, that the RWD is outperformed by other random walk strategies in the knowledge acquisition task [2, 13]. In other words, while the RWD is efficient in recovering nodes' properties, it takes longer to discover new nodes. Another interesting finding is that many of the real-world networks displayed a behavior that is similar to the one observed for the respective models. This is the case of the Facebook network, which displayed a behavior consistent with BA networks.

## 5 Conclusions

In the current paper, we proposed a framework to identify the efficiency in recovering network metrics arising from the reconstructed structure generated by a sequence of symbols. Networks were reconstructed using the well-known co-occurrence approach. The efficiency in recovering the network structure was evaluated by comparing reconstructed and original networks via correlation of network metrics. The analysis included four network models and six real networks. Five different random walks were evaluated. Here we focused on analyzing the ability to recover network metrics as many networked-based applications depend on the accurate representation of network topology [38, 39].

Our experiments revealed that long walks do not necessarily yield a high correlation between original and reconstructed networks. We also found that the TSAW Edge dynamics achieved a high correlation for most of the experiments. Surprisingly, while having a similar strategy to select neighbors, the TSAW Node dynamics did not achieve competitive correlations. In modular networks, the walking strategy based on avoiding hubs did not achieve competitive performance in particular network models (e.g. RWID for all considered values of mixing parameter). Conversely, the RWD dynamics, performed well in most scenarios, especially when the size of the sequence size used in the reconstructed network was typically lower than 2, 000 nodes. Such behavior was similar for model and real networks.

We also analyzed the interplay between network reconstruction and knowledge acquisition performance. The experiments demonstrated that the RWD outperforms other dynamics with regard to network metrics recovery efficiency. However, this random walk discovers new nodes slower than others. In other words, discovering nodes faster may not reflect in a good local network representation via network metrics. Finally, we also noted that the true self-avoiding walking—an efficient metric in the knowledge acquisition task—might have distinct behavior in recovering network metrics depending on which network elements are avoided. We found that avoiding visited edges is more efficient in network metrics recovery than avoiding nodes, according to the true self-avoiding rule.

In general, for shorter walks, the performance in recovering the properties of the original networks can vary substantially depending on its architecture and type of walk dynamics. In addition, for some combinations of networks, walks and metrics; even longer walks can lead to low correspondence between the real and observed properties. Such results indicate that biases can be easily formed depending on the walk dynamics, length of the sequences, and topological characteristics of the networks.

A potential application of this work is understanding how recommendation algorithms (in social media or content platforms) impact in the perceived knowledge of the network. For instance, we found that clustering coefficient was not reliably recovered from network reconstructions based on the RWD dynamics. This suggests that when new content is recommended to users based on their number of views (or number of links to them, similar to the RWD

dynamics), this may lead to the misleading notion that related contents (local) are not inter-connected among themselves but only through hubs.

Another application is understanding the different user behaviors in click-streams [40] data. Such a type of data covers sequences of web access or actions taken in by users in a online platform or across the whole internet. Users may navigate across content by using different strategies, which could potentially be identified by the patterns of the reconstructed networks. A similar approach could be used to understand the foraging process of researchers in science [41], i.e., the different strategies they use to perform or seek for new experiments, research questions and theories. This can be accomplished by considering researchers as agents walking across a knowledge space made from publications [42].

While this paper focused on a global recovery strategy, in future studies we intend to analyze whether different parts of the network are more easily recovered. In addition, we also intend to analyze if other reconstruction methods lead to improved reconstruction accuracy. The results could lead to potential new approaches to model sequences as complex networks, with potential implications in applications relying on co-occurrence approaches [43]. The proposed framework can also be used to better understand and aid developing new algorithms to estimate topological features of networks based on samples or partial observations of the data, such as in [44, 45]. In addition to that, future work can also focus on walks directly inspired by content recommendation strategies commonly used in media content platforms, which can lead to new ways to diversify content for the users or to mitigate biases in these platforms.

## Author Contributions

**Conceptualization:** Lucas Guerreiro, Filipi Nascimento Silva, Diego Raphael Amancio.

**Data curation:** Lucas Guerreiro, Filipi Nascimento Silva.

**Funding acquisition:** Diego Raphael Amancio.

**Investigation:** Lucas Guerreiro, Filipi Nascimento Silva.

**Methodology:** Lucas Guerreiro, Filipi Nascimento Silva.

**Project administration:** Filipi Nascimento Silva, Diego Raphael Amancio.

**Supervision:** Filipi Nascimento Silva, Diego Raphael Amancio.

**Visualization:** Lucas Guerreiro.

**Writing – original draft:** Lucas Guerreiro, Filipi Nascimento Silva, Diego Raphael Amancio.

**Writing – review & editing:** Lucas Guerreiro, Filipi Nascimento Silva, Diego Raphael Amancio.

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
