## [Decision Letter · Decision Letter 0]

17 Sep 2023

PONE-D-23-25848Identifying the Perceived Local Properties of Networks Reconstructed from Biased Random WalksPLOS ONE

Dear Dr. Silva,

Thank you for submitting your manuscript to PLOS ONE. After careful consideration, we feel that it has merit but does not fully meet PLOS ONE’s publication criteria as it currently stands. Therefore, we invite you to submit a revised version of the manuscript that addresses the points raised during the review process.

We look forward to receiving your revised manuscript.

Kind regards,

Dariusz Siudak, Ph.D., DSc.

Academic Editor

PLOS ONE

Journal Requirements:

"Diego R. Amancio acknowledges financial support from CNPq (grant no. 311074/2021-9) and São Paulo Research Foundation (FAPESP grant no. 2020/06271-0)."

"Diego R. Amancio acknowledges financial support from CNPq (grant no. 311074/2021-9) and São Paulo Research Foundation (FAPESP grant no. 2020/06271-0). The funders had no role in study design, data collection and analysis, decision to publish, or preparation of the manuscript."

Additional Editor Comments:

I propose to extend the assessment of the similarity of partitions in the original and reconstructed network using the ARI measure in addition to NMI.

Reviewers' comments:

Reviewer's Responses to Questions

**Comments to the Author**

1. Is the manuscript technically sound, and do the data support the conclusions?

Reviewer #1: Yes

Reviewer #2: Yes

2. Has the statistical analysis been performed appropriately and rigorously? 

Reviewer #1: Yes

Reviewer #2: Yes

3. Have the authors made all data underlying the findings in their manuscript fully available?

Reviewer #1: No

Reviewer #2: Yes

4. Is the manuscript presented in an intelligible fashion and written in standard English?

Reviewer #1: Yes

Reviewer #2: Yes

5. Review Comments to the Author

Reviewer #1: In their work entitled “Identifying the Perceived Local Properties of Networks Reconstructed from Biased Random Walks", the authors consider how to discover local properties of networks by using the reconstructed networks from biased random walks. Different walk strategies on different networks are addressed and the correlation of the local properties between the reconstructed networks and the original networks are evaulated.

The topic considered here is particularly meaningful and has wide applications, e.g. machine learning. I like this work and I think the results worth of publication potentially. However, the manuscript, in the current state, displays some flaws that the authors must address very carefully before I can recommend publication.

1. Is the data for the real networks such as Facebook and the power grid,and etc, active and traceable? Can the authors provide related download links?

2. In Sec. 3.3, the authors investigated the results for the following set of sequence length w. However, the size of the networks may have big effect on the correlation between the reconstructed networks and the original networks. How did the authors handle this issue?

3.The sequence of nodes obtained by random walk has a high degree of randomness. Different results will be obtained at different time. How did the author handle this issue?

4. To improve the readability, it is better to point out they use Pearson correlation to measure the correlation of the structural properties between the reconstructed networks and the original networks.

Reviewer #2: This manuscript focuses on network reconstruction to recover the local properties of networks generating sequences of symbols for different combinations of random walks and network topologies.

The experiments are conducted on four types of network models (i.e. Erdős-Rényi, Barabási-Albert, Waxman and Modular Networks) and six types of real-world networks (i.e. Facebook, Power Grid, Econ-Poli, Web-EPA, Bio (DM-CX), socfb-JohnsHopkins55). The considered network models have 5,000 nodes, and the real ones have from 320 to 5,180 nodes.

Then, five types of random walks (i.e. traditional random walk, degree-biased random walk, inverse of degree-biased random walk, true self-avoiding random walk on nodes, true self-avoiding random walk on edges) are used to explore the initial network and generate a sequence of visited nodes which is assumed to be the information available for network reconstruction.

Degree, clustering coefficient, closeness, betweenness eccentricity and coreness centrality of the networks are computed to analyze if relevant properties of the initial networks have been recovered.

In my opinion, the problem is certainly worth to be investigated. The results from the paper are significant and sufficient to be published in the journal.

Nevertheless, the used methodology should be expressed more clearly and logically. Maybe a pseudocode or a list of steps can be used to summarize the procedure used to prune and reconstruct the network, and also to compute the efficiency in recovering the network structure.

In the Related Works section, the authors refers to the use of random walkers exploring complex network topologies. At first, I wish to suggest the following references on a recent defined centrality measure named Game of Thieves which could be interesting for the authors’ future works:

- Ficara, A., Fiumara, G., De Meo, P., Liotta, A. (2021). Correlations Among Game of Thieves and Other Centrality Measures in Complex Networks. In: Fortino, G., Liotta, A., Gravina, R., Longheu, A. (eds) Data Science and Internet of Things. Internet of Things. Springer, Cham. https://doi.org/10.1007/978-3-030-67197-6_3

- Ficara, A., Saitta, R., Fiumara, G., De Meo, P., Liotta, A. (2021). Game of Thieves and WERW-Kpath: Two Novel Measures of Node and Edge Centrality for Mafia Networks. In: Teixeira, A.S., Pacheco, D., Oliveira, M., Barbosa, H., Gonçalves, B., Menezes, R. (eds) Complex Networks XII. CompleNet-Live 2021. Springer Proceedings in Complexity. Springer, Cham. https://doi.org/10.1007/978-3-030-81854-8_2

Then, I wish to suggest to add some additional references on network reconstruction.

At the end, the paper should be properly proofread checking acronym definition, grammar, punctuation, mechanics and other mistakes such missing spaces.

6. PLOS authors have the option to publish the peer review history of their article (what does this mean?). If published, this will include your full peer review and any attached files.

Reviewer #1: No

Reviewer #2: No

---

## [Author Response · Author response to Decision Letter 0]

27 Nov 2023

Dear Editor,

Please find attached a revised version of our manuscript “Identifying the Perceived Local Properties of Networks Reconstructed from Biased Random Walks”, in which we have taken onboard all the referee’s suggestions. Please find attached a response sheet to the referee’s comments.

Yours sincerely,

Additional Editor Comments:

I propose to extend the assessment of the similarity of partitions in the original and reconstructed network using the ARI measure in addition to NMI.

Response: We have conducted experiments using the ARI measure in addition to NMI. Therefore, we have updated Figures 4 and 5, and changed the text. The changes are highlighted in blue color.

Reviewer #1: In their work entitled “Identifying the Perceived Local Properties of Networks Reconstructed from Biased Random Walks", the authors consider how to discover local properties of networks by using the reconstructed networks from biased random walks. Different walk strategies on different networks are addressed and the correlation of the local properties between the reconstructed networks and the original networks are evaulated.

The topic considered here is particularly meaningful and has wide applications, e.g. machine learning. I like this work and I think the results worth of publication potentially. However, the manuscript, in the current state, displays some flaws that the authors must address very carefully before I can recommend publication.

1. Is the data for the real networks such as Facebook and the power grid,and etc, active and traceable? Can the authors provide related download links?

Response: Thank you for pointing it out. All the data is publicly available, and the links are already on the paper in form of citations; we have added text to clarify it in the current version. During this review we have noticed that, in the version we have submitted, the AI Interactions paragraph was not included (we apologize for that), and we have added one paragraph with information regarding this network.

2. In Sec. 3.3, the authors investigated the results for the following set of sequence length w. However, the size of the networks may have big effect on the correlation between the reconstructed networks and the original networks. How did the authors handle this issue?

Response: Indeed, we acknowledge that the size of a network can influence the coverage of a random walk for a fixed sequence length w. To address this, we varied the value of w in our experiments. In Figure 6, the x-axis represents network coverage, which helps mitigate the dependence of our results on network size. This methodological choice allows us to focus more on comparing the dynamics within each network rather than across networks of different sizes.

Furthermore, to minimize the impact of network size variability, we carefully selected the dataset so that all real-world networks, except for the Facebook network, are of similar sizes. The artificial models were also constructed to have a number of nodes equivalent to these real-world counterparts. In the case of the Facebook network, which is notably smaller, we observed that the Pearson correlation reaches 1 more rapidly. This observation aligns with the expected influence of network size on correlation measures. We have included a new paragraph in the methodology about this issue.

3.The sequence of nodes obtained by random walk has a high degree of randomness. Different results will be obtained at different time. How did the author handle this issue?

Response: Indeed, biases may emerge due to the random selection of starting nodes in each walk. To address this issue, we implemented a method of repeating the random walks multiple times for each network combination. Consequently, the Pearson correlation values depicted in our plots represent the mean of these repetitions. Our extensive testing revealed that augmenting the count of these realizations did not significantly alter the resultant curves. We have included additional details on section 3.4 in the revised manuscript to clarify this approach.

4. To improve the readability, it is better to point out they use Pearson correlation to measure the correlation of the structural properties between the reconstructed networks and the original networks.

Response: We have clarified in the “Introduction” and “Correlation analysis” sections that we adopted the Pearson correlation as our measure of quality for network reconstruction concerning observed properties. This choice was made because the Pearson correlation effectively captures the general similarity between measurements of the structural properties in the reconstructed and original networks. 

Reviewer #2: This manuscript focuses on network reconstruction to recover the local properties of networks generating sequences of symbols for different combinations of random walks and network topologies.

The experiments are conducted on four types of network models (i.e. Erdős-Rényi, Barabási-Albert, Waxman and Modular Networks) and six types of real-world networks (i.e. Facebook, Power Grid, Econ-Poli, Web-EPA, Bio (DM-CX), socfb-JohnsHopkins55). The considered network models have 5,000 nodes, and the real ones have from 320 to 5,180 nodes.

Then, five types of random walks (i.e. traditional random walk, degree-biased random walk, inverse of degree-biased random walk, true self-avoiding random walk on nodes, true self-avoiding random walk on edges) are used to explore the initial network and generate a sequence of visited nodes which is assumed to be the information available for network reconstruction.

Degree, clustering coefficient, closeness, betweenness eccentricity and coreness centrality of the networks are computed to analyze if relevant properties of the initial networks have been recovered.

In my opinion, the problem is certainly worth to be investigated. The results from the paper are significant and sufficient to be published in the journal.

Nevertheless, the used methodology should be expressed more clearly and logically. Maybe a pseudocode or a list of steps can be used to summarize the procedure used to prune and reconstruct the network, and also to compute the efficiency in recovering the network structure.

In the Related Works section, the authors refers to the use of random walkers exploring complex network topologies. At first, I wish to suggest the following references on a recent defined centrality measure named Game of Thieves which could be interesting for the authors’ future works:

- Ficara, A., Fiumara, G., De Meo, P., Liotta, A. (2021). Correlations Among Game of Thieves and Other Centrality Measures in Complex Networks. In: Fortino, G., Liotta, A., Gravina, R., Longheu, A. (eds) Data Science and Internet of Things. Internet of Things. Springer, Cham. https://doi.org/10.1007/978-3-030-67197-6_3

- Ficara, A., Saitta, R., Fiumara, G., De Meo, P., Liotta, A. (2021). Game of Thieves and WERW-Kpath: Two Novel Measures of Node and Edge Centrality for Mafia Networks. In: Teixeira, A.S., Pacheco, D., Oliveira, M., Barbosa, H., Gonçalves, B., Menezes, R. (eds) Complex Networks XII. CompleNet-Live 2021. Springer Proceedings in Complexity. Springer, Cham. https://doi.org/10.1007/978-3-030-81854-8_2

Then, I wish to suggest to add some additional references on network reconstruction.

At the end, the paper should be properly proofread checking acronym definition, grammar, punctuation, mechanics and other mistakes such missing spaces.

Response: Thanks for the positive comments. Regarding the pseudocode, it was inserted under Section 3.4. We have included the suggested references and discussed them in the conclusion, in particular for the general problem of using sampled information from networks to estimate topological measures. We have proofread the document to check for problems.

---

## [Decision Letter · Decision Letter 1]

6 Dec 2023

Identifying the Perceived Local Properties of Networks Reconstructed from Biased Random Walks

PONE-D-23-25848R1

Dear Dr. Silva,

We’re pleased to inform you that your manuscript has been judged scientifically suitable for publication and will be formally accepted for publication once it meets all outstanding technical requirements.

Kind regards,

Dariusz Siudak, Ph.D., DSc.

Academic Editor

PLOS ONE

Additional Editor Comments (optional):

Reviewers' comments:

Reviewer's Responses to Questions

**Comments to the Author**

1. If the authors have adequately addressed your comments raised in a previous round of review and you feel that this manuscript is now acceptable for publication, you may indicate that here to bypass the “Comments to the Author” section, enter your conflict of interest statement in the “Confidential to Editor” section, and submit your "Accept" recommendation.

Reviewer #1: All comments have been addressed

Reviewer #2: All comments have been addressed

2. Is the manuscript technically sound, and do the data support the conclusions?

Reviewer #1: Yes

Reviewer #2: Yes

3. Has the statistical analysis been performed appropriately and rigorously? 

Reviewer #1: Yes

Reviewer #2: Yes

4. Have the authors made all data underlying the findings in their manuscript fully available?

Reviewer #1: Yes

Reviewer #2: Yes

5. Is the manuscript presented in an intelligible fashion and written in standard English?

Reviewer #1: Yes

Reviewer #2: Yes

6. Review Comments to the Author

Reviewer #1: The topic considered here is particularly meaningful and has wide applications, e.g. machine learning. I like this work and I think the results worth of publication. For the points raised in the previous comment, all my concerns have been addressed. I thus recommend publication as is.

Reviewer #2: (No Response)

7. PLOS authors have the option to publish the peer review history of their article (what does this mean?). If published, this will include your full peer review and any attached files.

Reviewer #1: No

Reviewer #2: No

---

## [Editor Report · Acceptance letter]

11 Jan 2024

PONE-D-23-25848R1 

PLOS ONE

Dear Dr. Silva, 

I'm pleased to inform you that your manuscript has been deemed suitable for publication in PLOS ONE. Congratulations! Your manuscript is now being handed over to our production team.

Kind regards, 

on behalf of

Dr. Dariusz Siudak 

Academic Editor

PLOS ONE